Unveiling the impact of organically activated biochar on physiological, biochemical, and yield attributes of maize under varied field moisture conditions

Iqbal Hajra 1
Jahan Summera summera.botany@pu.edu.pk 1 2
Akbar Muhammad 1
Kamran Atif 2 3
Rasul Fahd 4
Shahzad Sobia 5
Siddiqui Manzer H. 6
Alamri Saud 6
1 Department of Botany, University of Gujrat , Gujrat , Punjab , Pakistan
2 Institute of Botany, University of the Punjab , Lahore , Punjab , Pakistan
3 Agricultural Food and Nutritional Science, University of Alberta , Edmonton , Canada
4 Department of Agronomy, University of Agriculture Faisalabad , Faisalabad , Punjab , Pakistan
5 Department of Botany, Islamia University Bahawalpur , Bahawalnagar Campus , Punjab , Pakistan
6 Department of Botany and Microbiology, King Saud University , Riyadh , Saudi Arabia
Serim Ahmet Tansel
Electronic publication date: 2024 Oct 1
Publication date: 2024
Volume: 12
Electronic Location ID: e17883
Received 2024 May 21; Accepted 2024 Jul 17
Copyright: ©2024 Iqbal et al.
Copyright year: 2024
Copyright holder: Iqbal et al.
License: This is an open access article distributed under the terms of the Creative Commons Attribution License, which permits unrestricted use, distribution, reproduction and adaptation in any medium and for any purpose provided that it is properly attributed. For attribution, the original author(s), title, publication source (PeerJ) and either DOI or URL of the article must be cited.
License URL: https://creativecommons.org/licenses/by/4.0/

Keywords: Nutrient composition of soil, Water stress, FTIR analysis of maize metabolites, Photosynthetic pigments, Yield attributes

Funding: Higher Education Commission (HEC) of Pakistan under NRPU (National Research Program for Universities) Project 20-16716 King Saud University, Riyadh, Saudi Arabia RSP2024R194 The study was supported by Higher Education Commission (HEC) of Pakistan under NRPU (National Research Program for Universities) Project, grant No. 20-16716 and also the Researchers Supporting Project number (RSP2024R194), King Saud University, Riyadh, Saudi Arabia. The funders had no role in study design, data collection and analysis, decision to publish, or preparation of the manuscript.

==============================
Water deficiency in semiarid regions is a limiting factor that affects crop quality and yield. In Punjab, Pakistan, a 27% decline in maize yield was detected over the past two decades just because of water scarcity. Currently, no studies have reported the effects of organically activated biochar (AB) on crop productivity under natural field conditions. For this purpose, a field experiment in a split-split-plot design was conducted with three amendment levels (0, 2, and 4 tonnes ac−1), and three maize hybrids (DK-9108, DK-6321, and Sarhaab) under 100%, 70%, and 50% irrigation water (IW) of crop evapotranspiration (ETc). The AB significantly improved the soil’s physical and chemical properties, and maximum improvement was recorded in 4 tonnes ac−1 AB amendment in organic matter (16.6%), total organic carbon (17%), phosphorus (11.43%), and available potassium (29.27%). The 4 tonnes ac−1AB amendment in soil had a significant impact on total chlorophyll content (0.3–1-fold in DK-6321), carotenoid content (3.9–4.4-fold in Sarhaab), and relative water content (30% and 21% in Sarhaab) under 50% irrigation water (IW) of ETc at V14th and R3 stages, respectively. Moreover, a significant decline in stress markers (proline content and sugar content) was detected at both growth stages in all maize hybrids in AB amended soil. The analysis of plant metabolites indicated increased intensities of phenolics, alkyl esters, and carbohydrates by 2.5–7%, 17–80%, and 40–43% in DK-6321 under 50% IW in 2–4 tonnes ac−1 AB amended soil, respectively. The highest improvement in growth and yield attributes among maize hybrids was detected in the order DK-6321¿DK-9108¿Sarhaab in 2–4 tonnes ac−1AB amended soil under 70% and 50% IW of ETc, respectively. Hence, this research might help to develop an effective soil amendment to restore degraded soils and improve maize growth under arid climatic conditions.

Introduction

The current climatic patterns indicate that increasing temperatures, inadequate rainfall, and severe drought conditions will continue to escalate in the near future (Naumann et al., 2014). Recent studies have revealed that more severe, frequent, prolonged, and widespread drought spells will occur due to global warming (Chiang, Mazdiyasni & AghaKouchak, 2021). The reports of the world bank group indicate that drought will affect over 1.5 to 1.7 billion people in South Asia by the year 2050 (Sheoran et al., 2022). Various risk assessments indicate that arid and semi-arid regions with less developed economies and heavy reliance on agriculture will face an intensified risk of drought (Wang & Sun, 2023). Moreover, the soil on the main farming land has undergone deterioration, leading to inadequate water retention (with water in the saturation zone remaining below a depth of 1.5 m) and decreased capacity to retain nutrients (Grote et al., 2021). This degradation can be attributed to excessive agricultural practices, which in turn have led to a decline in crop yields and an increased reliance on agricultural imports (Derpsch et al., 2024; Rawls et al., 2003). Maize is a significant cash crop that plays a substantial role in agricultural food security (Poole, Donovan & Erenstein, 2021). The global cultivation area for maize spans approximately 197 million hectares, encompassing most parts of Asia, sub-Saharan Africa, and Latin America (FAOStat, 2021). The economic survey of Pakistan for the years 2021–22 indicates that the cultivation of the maize crop is practiced on an extensive area of 1,653 thousand hectares, resulting in a substantial yield of approximately 10.635 million metric tonnes that accounts for 3.2% of total agricultural output and 0.7% of GDP. However, inadequate moisture throughout critical growth phases is lowering maize yields (Sah et al., 2020). It hampers growth and development by triggering alterations in various physiological and biochemical constituents as it disrupts chlorophyll synthesis, speeds up chlorophyll breakdown, and hinders photosynthesis (Széles et al., 2023). Earlier research work suggested that water constraints could result in a considerable drop in maize output, potentially up to 20%, 50%, and 21% at vegetative, flowering, and grain filling stages, respectively (Sheoran et al., 2022).

To address the issue of lower yields of maize crops due to a lack of appropriate moisture during critical growth phases, it is necessary to develop an effective strategy, especially for small-scale farmers, that can rehabilitate soil conditions (by improving soil hydrology and fertility) and maize productivity (Rehrah et al., 2016). Organic soil conditioners, such as compost, sawdust, reeds, and manures, are generally regarded as preferable alternatives due to their high organic content, which improves the soil’s ability to retain nutrients and water (Santana et al., 2018; Williams & Thomas, 2023). Although these organic amendments add nutrients to the soil, but the benefits are usually temporary. This is due to the fact that nutrients tend to leach down into groundwater, and organic matter undergoes rapid decomposition into CO2, particularly in semiarid regions (Bhatt & Hossain, 2019).

Biochar has been recognized as the most effective approach for achieving long-term soil sustainability (Fang et al., 2018; Wu et al., 2017). Recently, there has been a huge interest worldwide in the use of biochar to increase soil fertility (Chen et al., 2015; Oliveira et al., 2017). Biochar is produced by the pyrolysis of waste biomass above 250 °C in an oxygen-free environment, and the resultant product is very porous, carbon-rich, and resilient to decay (Pandey, 2023), and can stay for long periods in soil due to its distinctively resilient aromatic structure (Biederman & Harpole, 2013). Amending soil with biochar can affect its water retention characteristics by reducing soil bulk density, enhancing total soil pore volume, changing pore-size distribution, increasing soil surface area (particularly in coarse-textured soils), and improving soil aggregation (Wang et al., 2019). A variety of organic resources can be utilized to produce biochar, such as wood, fruit shells, chicken manure, agricultural residue, food waste, biosolids, and wood (Ippolito et al., 2020; Tan & Lee, 2016). However, wood-derived biochar has outstanding properties, such as a substantial and stable aromatic carbon structure (Sánchez-Monedero et al., 2019), which makes it a more suitable soil conditioner to hold water for longer durations (Jahan et al., 2019). The highest lignin, hemicellulose, and cellulose content of acacia wood is more advantageous because, upon pyrolysis, it transforms into a recalcitrant carbon structure (Pituya, Sriburi & Wijitkosum, 2017; Shebani, Van Reenen & Meincken, 2009). Slow pyrolysis preserves the wood’s structure while enhancing pore diameter, pore volume, and absorption capacity (Fryda & Visser, 2015). Acacia wood is abundant in South Asia; therefore, using its biochar as a soil conditioner in agricultural production is rational and affordable (Shakeel et al., 2022). The highly porous structure of biochar holds a sufficient quantity of capillary water inside its micropores for longer durations (Jahan et al., 2019), which becomes available to plants under water scarce conditions; hence, biochar has the potential to mitigate the adverse effects of drought (Narzari et al., 2017; Xiang et al., 2023). The surface area of biochar can be enhanced by its activation, which involves the improvement of its surface through organic or chemical treatments (Tan & Lee, 2016). However, organic activation of biochar with vermicompost and perlite is the most cost-effective and ecologically friendly technique. This study introduces the innovative concept of applying organically activated biochar (AB) to soil, with the hypothesis that it will enhance soil water retention capacity and improve maize growth under deficit irrigation. By reclaiming and reusing essential nutrients and bioenergy from organic materials for biochar’s activation, this method decreases reliance on chemicals as it embraces agroecological farming practices, which are gaining worldwide interest as appealing and safer alternatives to chemicals in food production (Becagli, Arduini & Cardelli, 2022). Previous work reported an increase of 2.5-folds in surface area and 17-folds in pore volume of organically activated biochar over fresh biochar, which ultimately caused an improvement in water holding capacity of a loamy soil from 31% to 41% with 2.5% and 5% amendment rates, respectively (Jahan et al., 2023). At present, most research work has been devoted to the physical and chemical activation of biochar, as well as small-scale pot experiments, and none of the studies have reported the influence of organically activated biochar on crop productivity in natural field conditions under water stress. Hence, the current study aimed to find out the influence of organically activated biochar on soil physicochemical attributes as well as physio-biochemical, secondary metabolites, and yield attributes of three maize genotypes with three amendment levels of activated biochar under three different moisture regimes. This research will be instrumental in developing effective soil amendments to restore degraded soils, thereby enhancing its water retention capacity. Such advancements will support the achievement of Sustainable Development Goals, particularly by promoting a sustainable food system and improving livelihoods.

Material and Methods

Preparation, activation and amendment of biochar in soil

Biochar production was undertaken in 2022 by utilizing raw material sourced from the gum Arabica tree (Acacia nilotica) at the University of Agriculture, Faisalabad. Following a drying phase, acacia shavings were subjected to pyrolysis at a temperature of 450 °C for a duration of 3 h. For the preparation of vermicompost, corn stover was employed as the raw material, and the red earthworm species Eisenia fetida was employed for the vermicomposting process. Biochar was activated according to the procedure of Jahan et al. (2023). Three concentrations 0 tonnes ac−1, 2 tonnes ac−1 and 4 tonnes ac−1 of activated biochar was amended by thorough mixing in the top 15 cm of soil. The experimental layout was consisted of split-split-plot design. After three month of amending activated biochar in the field soil sample were acquired for physicochemical testing.

Soil analysis

Three soil samples were taken from from the top 0–20 cm layer from each treatment and a representative sample was attained by thorough mixing of sub-samples. After heating in an oven at 80 °C for 24 h, the samples were ground and sieved for further testing. The loamy sand texture of the soil was ascertained by the Brady (1990) procedure. The estimation of pH values was made by following the standard procedure ISO 10390. The electrical conductivity of the soil was calculated by the method given by Rayment & Higginson (1992). Soil organic matter, oxidizable organic carbon, total organic carbon was determined by the method of George et al. (2017). While, soil phosphorus was determined by the procedure of Olsen & Sommers (1982). Soil potassium was determined by flame photometer.

Field experiment

The field experiment was conducted at location 32°38.37′N, 74°9.00′E (Gujrat) for exploring the suitable level of activated biochar under three moisture regimes (100% ETC, 70% ETC, and 50% ETC). The field was chosen on account of its low water holding capacity (18% FC) due to loamy sand soil and arid climatic conditions. The following seeds of maize hybrids was used DK-9108, DK-6321 and Sarhaab.

The experiment was placed out in a split-split-plot design with three replications. Healthy seed of three maize varieties were sown in subplots on 19th February, 2023. There were 27 treatments with three replicates. The area of each treatment plot was 16 m2; width = 2 m and length = 5.33 m, each having 3 rows with 0.3-meter row spacing. The following treatments were applied: T1 = DK-9108 + 0 tonnes ac−1 AB + 100% irrigation water (IW) of FC, T2 = DK-9108 + 0 tonnes ac−1 AB + 70% IW, T3 = DK-9108 + 0 tonnes ac−1 AB + 50% IW, T4 = DK-9108 + 2 tonnes ac−1 AB + 100% IW, T5 = DK-9108 + 2 tonnes ac−1 AB + 70% IW, T6 = DK-9108 + 2 tonnes ac−1 AB + 50% IW, T7 = DK-9108 + 4 tonnes ac−1 AB + 100% IW, T8 = DK-9108 + 4 tonnes ac−1 AB + 70% IW, T9 = DK-9108 + 4 tonnes ac−1 AB + 50% IW, T10 = DK-6321 + 0 tonnes ac−1 AB + 100% IW, T11 = DK-6321 + 0 tonnes ac−1 AB + 70% IW, T12 = DK-6321 + 0 tonnes ac−1 AB + 50% IW, T13 = DK-6321 + 2 tonnes ac−1 AB + 100% IW, T14 = DK-6321+ 2 tonnes ac−1 AB + 70% IW, T15 = DK-6321 + 2 tonnes ac−1 AB + 50% IW, T16 = DK-6321 + 4 tonnes ac−1 AB + 100% IW, T17 = DK-6321 + 4 tonnes ac−1 AB + 70% IW, T18 = DK-6321 + 4 tonnes ac−1 AB + 50% IW, T19 = Sarhaab + 0 tonnes ac−1 AB + 100% IW, T20 = Sarhaab + 0 tonnes ac−1 AB + 70% IW, T21 = Sarhaab + 0 tonnes ac−1 AB + 50% IW, T22 = Sarhaab + 2 tonnes ac−1 AB + 100% IW, T23 = Sarhaab + 2 tonnes ac−1 AB + 70% IW, T24 = Sarhaab + 2 tonnes ac−1 AB + 50% IW, T25 = Sarhaab + 4 tonnes ac−1 AB + 100% IW, T26 = Sarhaab + 4 tonnes ac−1 AB + 70% IW, T27 = Sarhaab + 4 tonnes ac−1 AB + 50% IW.

Soil moisture was maintained by Eq. (1) (Faghih et al., 2021). (1) I=ETc=θfc−θiDR

where, “I” represents the total of irrigation water (mm), “θfc” indicates volumetric moisture content at field capacity and “θi” represents the soil moisture content before irrigation. While, “DR” indicates the root depth.

The volumetric moisture content at field capacity of soil before (θfc) and after (θi) irrigation was monitored by soil moisture meter PMS-714. The field capacity of the soil was determined by assessing the moisture content 72 h after fully saturating the soil while covered from the top to avoid evaporation.

Plant sampling and analysis

For estimation of physiological and biochemical attributes of maize under different treatments, fresh leaves were sampled at V14 (collar of 14th leaf become visible) stage and R3 (milking stage) on 9th April, 2023 and 19th May, 2023, respectively. The leaf relative water content was calculated by applying the Eq. (2) according to Weatherley’s (1950) method. (2) Leafrelativewatercontent=Freshweight−DryweightSaturatedweight−Dryweight×100.

The chlorophyll and carotenoid pigment profiling of maize leaves was done by the method of Arnon (1949) and Lichtenthaler & Wellburn (1983), respectively. Crushing of leaf sample (0.1 g) was done with 5 mL of 80% acetone. The solution was centrifuged (15,000×g for 5 min) and the upper layer was transferred to the clean cuvette of the spectrophotometer (UV-Vis; Shmadzu 35, RP China) and absorbance was recorded at 663-nm, 645-nm and 470-nm.

Leaf sugar contents were estimated by crushing 0.5 g leaf material in clean mortar with (10 mL) distilled water (DuBois et al., 1956). In a clean test tube 0.1 mL of filtrate with 1 mL of phenol (5% v/v) was added. After the incubation at room-temperature for 60 min, 5 mL of concentrated H2SO4 was added in the test tube. The same procedure was repeated for the blank and filtrate from other treatments. The absorbance value of the sample and blank was noted at 420 nm by using spectrophotometer.

The leaf proline content was determined by the method of Bates, Waldren & Teare (1973). By following this procedure, pre-chilled dry leaf 0.1 g was ground in a mortar after that homogenized it with 4 mL sulfosalicylic acid (3%). This mixture was kept at 5 °C for 24 h. Then the extract was centrifuged at 3000 rpm for 5 min. Acetic ninhydrin reagent (4 mL) was mixed with supernatant and after properly shaking the mixture; the test tube was placed for one hour at 100 °C in the water-bath. After cooling the extract toluene (4 mL) was added in it. The absorbance of the toluene layer was analyzed at 520-nm in a spectrophotometer.

The FT-IR spectra of powdered leaf samples were obtained using the KBr (potassium bromide) method, utilizing a Nicolet iS5 Thermo Scientific instrument (USA), following the procedure outlined by Celi, Schnitzer & Nègre (1997). The infra-red spectra of the leaf samples were obtained in the range of 4000 to 500 cm−1. Further processing and analysis of the spectra were conducted using Origin Pro 2020.

Plants were collected at maturity in triplicates for absolute growth analysis. Shoot, leaves and roots of the plants were dissected properly and their fresh weight was calculated in gram by using the digital electronic balance. After that, roots, stems and leaves were placed on the chart and scanned. Further processing on the scanned images was done by Image J software (NHIV, v. 1.51 64-bit, Bethesda, MD, USA) and the length of stem, root and leaf area was measured. The counting of number of leaves per plant was done manually. The dry weights of the stem, roots and leaves were recorded after oven drying the sample for 72 h at 70 °C. At maturity following yield attributes were determined; ear length (cm), 1,000 grain weight, number of grains rows per ear, grain weight per ear.

Statistical analysis of data

The MINITAB 19 software was used to apply the general linear model (GLM) for univariate analysis. The significance of the treatment was determined using the Tukey’s least significant difference (LSD) test at a 95% confidence level. The chord diagram of soil physio-chemical attributes and heatmap of plant growth parameters were made by using R software (R−4.3.1-x86 64.pkg).

Results

Soil analysis

The interaction between levels of AB amendment (0, 2, and 4 tonnes acre−1) and soil physicochemical properties is presented in the form of a chord diagram (Fig. 1). The chord diagram shows the relation between AB amendment level (at the base of the circle) and associated soil characteristics (pH, electrical conductivity (EC), organic matter, available phosphorus, available potassium, organic carbon, and oxidizable organic carbon). Each chord connecting an AB amendment level to soil characteristics, i.e., EC, pH, and organic matter, highlights the strength of association, with thicker chords indicating stronger association. The chord diagram indicates a strong association between soil amended with AB at rates of 2 and 4 tonnes acre−1 and the EC, which was significantly (P < 0.05) higher at 4.4% and 7.3%, respectively, than that of soil not amended with AB. In contrast to soil without AB amendment, 4 tonnes acre−1 biochar-amended soil exhibited a significant decrease (4.34%) in pH (P < 0.05). The application of 2- and 4-tonnes acre−1 organically activated biochar resulted in a substantial (P < 0.05) increase of 7.14% and 16.6% in soil organic matter respectively. Findings showed that adding 2 and 4 tonnes of AB to the soil improved the available phosphorous and potassium by 9.72 to 11.43% and 9.96 to 29.27%, respectively. Additionally, soil total organic carbon increased, showing a significant difference of 8.33% at the 2 tonnes acre−1 amendment level and 16.6% at the 4 tonnes acre−1 amendment level. Similarly, organically activated biochar amendment at 2 and 4 tonnes acre−1 significantly enhanced oxidizable carbon content in soil by 7.18% and 17.68%, respectively, compared to non-amended soil.

Figure 1 Influence of 0 tonnes acre−1(TA), 2 tonnes acre−1, 4 tonnes acre−1 activated biochar amendment on EC (electrical conductivity), pH, OM (organic matter), available phosphorus (P mg kg−1), available potassium (k; mg kg−1), nitrogen (N; %), total organic carbon (TOC; %), oxidizable carbon (OC; %).

Physiological and biochemical analysis

Photosynthetic pigments

The analysis of variance (ANOVA) was used to compare the interaction of photosynthetic pigments among irrigation water (IW) of 100%, 70%, and 50% of ETc, amendment levels (0, 2, 4 tonnes acre−1), and three maize varieties (DK-6321, DK-9108, and Sarhaab) at two growth stages (Table 1). The results revealed that, at the V14th stage, DK-9108 had the lowest chlorophyll a content in soil without AB amendment under 50% IW of ETc. While at the R3 stage, DK-6321 had the lowest chlorophyll a content in soil without AB amendment. However, when treated with 4-tonnes acre−1 of AB amendment under 70% IW, DK-6321 exhibited significantly (P < 0.05) improved chlorophyll a content at V14th stage (14.74%) and R3 stage (1.5 folds), compared to non-amended soil. Water stress reduced chlorophyll b content in the leaf during both the V14th and R3 stages in all maize hybrids, with Sarhaab having the lowest chlorophyll b content in soil without AB amendment under 50% IW. Water stress (50% IW) at both growth stages reduced total chlorophyll content in all maize hybrids, with Sarhaab having the lowest total chlorophyll content at the V14th stage and DK-6321 having the lowest at the R3 stage. On the other hand, DK-6321 underwent significant improvement (60%–234% higher) in total chlorophyll content in 4-tonnes acre−1 AB amended soil under 70% IW at V14th and R3 stages, respectively. At the V14th stage, DK-9108 had a significantly higher carotenoid content (25.9%) in 4-tonnes acre−1 AB amended soil under 100% ETc compared to DK-9108 in 0-tonnes acre−1 AB amended soil. Similar to this, plants grown in 70% IW in DK-9108 during the R3 stage had 17.5% more carotenoids in 4-tonnes acre−1 AB-amended soil than plants grown in 70% IW without AB-amended soil. On the other hand, Sarhaab had significantly (P < 0.05) lower levels of carotenoids at the V14th stage (10.5%) and R3 stage (8.74%) when the soil was amended with AB under 50% IW compared to plants in soil without AB amendment under 100% IW.

Table 1 The impact of organically activated biochar on chlorophyll a, chlorophyll b, total chlorophyll, and carotenoid content at V14 (fourteen leaf stage) and R3 (milking) stages of maize under different moisture conditions.

Treatment	V14	R3	V14	R3	V14	R3	V14	R3	
	Chlorophyll a	Chlorophyll b	Total Chlorophyll	Carotenoid Content	
V 1 ×T 0 ×100%IW	2.68 ± 0.06 (j-n)	3.11 ± 0.04 (n)	0.782 ± 0.14 (c-d)	0.947 ± 0.06 (b-c)	3.46 ± 0.12 (de)	4.05 ± 0.24 (g)	0.616 ± 0.02 (cd)	0.869 ± 0.1 (b-c)	
V 1 ×T 0 ×70%IW	2.49 ± 0.33 (n-o)	3.37 ± 0.01 (m)	0.749 ± 0.2 (d)	0.847 ± 0.02 (h)	3.24 ± 0.92 (e)	4.21 ± 0.08 (g)	0.581 ± 0.03 (e)	0.778 ± 0.04 (h-i)	
V 1 ×T 0 ×50%IW	2.29 ± 0.09 (o)	2.69 ± 0.03 (o)	0.4 ± 0.093 (f)	0.754 ± 0.008 (i)	2.69 ± 0.34 (fg)	3.45 ± 0.03 (i)	0.56 ± 0.17 (e)	0.69 ± 0.01 (j)	
V 1 ×T 1 ×100%IW	2.78 ± 0.003 (g-m)	4.58 ± 0.01 (f-h)	1.024 ± 0.005 (a)	0.962 ± 0.003 (a-b)	3.81 ± 0.02 (c)	5.55 ± 0.01 (d)	0.777 ± 0.008 (a)	0.872 ± 0.01 (b-c)	
V 1 ×T 1 ×70%IW	2.83 ± 0.09 (f-m)	4.7 ± 0.01 (e)	0.95 ± 0.008 (a-c)	1.001 ± 0.003 (a)	3.78 ± 0.06 (c)	5.7 ± 0.013 (c)	0.723 ± 0.015 (ab)	0.891 ± 0.02 (a-b)	
V 1 ×T 1 ×50%IW	2.96 ± 0.18 (c-j)	4.49 ± 0.02 (h-i)	0.956 ± 0.023 (a-b)	0.95 ± 0.01 (b-c)	3.92 ± 0.03 (b)	5.44 ± 0.04 (e)	0.719 ± 0.03 (b)	0.843 ± 0.01 (c-f)	
V 1 ×T 2 ×100%IW	2.94 ± 0.06 (c-k)	4.84 ± 0.04 (d)	0.984 ± 0.028 (a)	0.947 ± 0.005 (b-c)	3.93 ± 0.02 (b)	5.79 ± 0.12 (bc)	0.748 ± 0.05 (a)	0.865 ± 0.009 (b-d)	
V 1 ×T 2 ×70%IW	2.97 ± 0.03 (b-i)	4.9 ± 0.02 (d)	0.939 ± 0.013 (a-c)	1.002 ± 0.01 (a)	3.91 ± 0.01 (b)	5.91 ± 0.03 (b)	0.715 ± 0.09 (b)	0.914 ± 0.01 (a)	
V 1 ×T 2 ×50%IW	2.71 ± 0.06 (h)	4.85 ± 0.10 (d)	0.868 ± 0.008 (a-d)	0.895 ± 0.008 (d-g)	3.58 ± 0.11 (d)	5.74 ± 0.02 (c)	0.662 ± 0.16 (c)	0.819 ± 0.01 (e-g)	
V 2 ×T 0 ×100%IW	2.69 ± 0.12 (i-n)	2.26 ± 0.01 (p)	0.894 ± 0.15 (b-d)	0.762 ± 0.005 (i)	3.59 ± 0.07 (d)	3.02 ± 0.02 (j)	0.44 ± 0.06 (f)	0.689 ± 0.006 (j)	
V 2 ×T 0 ×70%IW	2.85 ± 0.006 (e-l)	2.13 ± 0.03(q)	0.510 ± 0.08 (e)	0.858 ± 0.003 (g-h)	3.36 ± 0.01 (e)	2.99 ± 0.01 (k)	0.246 ± 0.03 (g)	0.775 ± 0.003 (i)	
V 2 ×T 0 ×50%IW	2.62 ± 0.06 (l-n)	2.04 ± 0.01 (q)	0.277 ± 0.01 (g)	0.79 ± 0.003 (i)	2.9 ± 0.09 (f)	2.83 ± 0.01 (l)	0.136 ± 0.006 (h)	0.717 ± 0.001 (j)	
V 2 ×T 1 ×100%IW	2.96 ± 0.03 (c-j)	4.58 ± 0.01 (f-h)	0.972 ± 0.028(a-b)	0.95 ± 0.003 (b-c)	3.93 ± 0.04 (b)	5.53 ± 0.03 (d)	0.738 ± 0.04 (a)	0.841 ± 0.004 (c-f)	
V 2 ×T 1 ×70%IW	3.25 ± 0.009 (a-b)	4.65 ± 0.01 (e-f)	0.873 ± 0.002 (a-d)	0.958 ± 0.003 (a-b)	4.12 ± 0.01 (a)	5.61 ± 0.01 (d)	0.665 ± 0.007 (c)	0.848 ± 0.002 (c-e)	
V 2 ×T 1 ×50%IW	2.83 ± 0.006 (f-m)	4.52 ± 0.01 (g-i)	0.754 ± 0.003 (d)	0.935 ± 0.005 (c-d)	3.58 ± 0.09 (d)	5.46 ± 0.02 (e)	0.575 ± 0.06 (e)	0.828 ± 0.01 (d-g)	
V 2 ×T 2 ×100%IW	3.22 ± 0.027 (a-c)	5.24 ± 0.02 (b)	0.914 ± 0.028 (a-d)	0.875 ± 0.003 (f-h)	4.13 ± 0.03 (a)	6.11 ± 0.01 (a)	0.694 ± 0.04 (bc)	0.803 ± 0.002 (f-i)	
V 2 ×T 2 ×70%IW	3.28 ± 0.042 (a)	5.382 ± 0.04 (a)	0.949 ± 0.03 (a-c)	0.889 ± 0.02 (e-h)	4.23 ± 0.09 (a)	6.27 ± 0.02 (a)	0.722 ± 0.09 (ab)	0.815 ± 0.01 (e-h)	
V 2 ×T 2 ×50%IW	2.91 ± 0.018 (d-k)	5.13 ± 0.04 (c)	0.872 ± 0.09 (a-d)	0.9 ± 0.02 (g-h)	3.79 ± 0.1 (c)	5.95 ± 0.02 (b)	0.664 ± 0.04 (c)	0.781 ± 0.003 (g-i)	
V 3 ×T 0 ×100%IW	2.66 ± 0.09 (k-n)	3.59 ± 0.01 (l)	0.871 ± 0.14 (f)	0.621 ± 0.002 (j)	3.53 ± 0.09 (d)	4.22 ± 0.002 (g)	0.423 ± 0.06 (f)	0.572 ± 0.001 (k)	
V 3 ×T 0 ×70%IW	2.8 ± 0.006 (g-m)	3.63 ± 0.01 (l)	0.524 ± 0.03 (f)	0.572 ± 0.05 (k)	3.32 ± 0.10 (e)	4.2 ± 0.04 (g)	0.261 ± 0.01 (g)	0.524 ± 0.004 (l)	
V 3 ×T 0 ×50%IW	2.56 ± 0.033 (m-o)	3.43 ± 0.02 (m)	0.233 ± 0.03 (f)	0.565 ± 0.04 (k)	2.79 ± 0.02 (f)	3.99 ± 0.04 (h)	0.125 ± 0.01 (h)	0.522 ± 0.002 (l)	
V 3 ×T 1 ×100%IW	3.14 ± 0.015 (a-d)	4.34 ± 0.06 (k)	1.001 ± 0.12 (a)	0.924 ± 0.03 (b-e)	4.14 ± 0.02 (a)	5.27 ± 0.04 (f)	0.758 ± 0.05 (a)	0.853 ± 0.01 (b-e)	
V 3 ×T 1 ×70%IW	3.2 ± 0.003 (a-d)	4.46 ± 0.02 (i-j)	0.912 ± 0.03 (a-d)	0.908 ± 0.03 (c-f)	4.11 ± 0.10 (a)	5.36 ± 0.03 (f)	0.692 ± 0.01 (c)	0.834 ± 0.02 (c-f)	
V 3 ×T 1 ×50%IW	3 ± 0.009 (a-h)	4.34 ± 0.01 (k)	0.804 ± 0.02 (b-d)	0.893 ± 0.03 (d-g)	3.8 ± 0.02 (c)	5.23 ± 0.04 (f)	0.613 ± 0.07 (d)	0.82 ± 0.01 (e-g)	
V 3 ×T 2 ×100%IW	3.12 ± 0.006 (a-e)	4.53 ± 0.01 (g-i)	0.945 ± 0.05 (a-c)	0.889 ± 0.05 (e-h)	4.07 ± 0.01 (ab)	5.42 ± 0.05 (e)	0.716 ± 0.02 (b)	0.82 ± 0.02 (e-g)	
V 3 ×T 2 ×70%IW	3.09 ± 0.012 (a-f)	4.6 ± 0.01 (f-g)	0.924 ± 0.02 (a-c)	0.902 ± 0.01 (d-g)	4.01 ± 0.05 (ab)	5.5 ± 0.02 (e)	0.701 ± 0.01 (b)	0.815 ± 0.004 (e-g)	
V 3 ×T 2 ×50%IW	3.01 ± 0.015 (a-g)	4.34 ± 0.06 (k)	0.891 ± 0.02 (b-d)	0.895 ± 0.03 (d-g)	3.9 ± 0.02 (b)	5.38 ± 0.03 (e)	0.677 ± 0.09 (c)	0.798 ± 0.01 (e-i)	
Notes.

The results represent the means derived from a three-way analysis of variance (ANOVA) involving maize varieties (V1= DK-9108, V22= DK-6321, and V3= Sarhaab) activated biochar amendment levels (T0= 0 tonnes acre−1, T1= 2 tonnes acre−1, T2= 4 tonnes acre−1), and irrigation water levels (100% IW= 100% irrigation water of crop evapotranspiration, expressed as evapotranspiration, 70% IW= 70% irrigation water of crop evapotranspiration, expressed as evapotranspiration, 50% IW= 100% irrigation water of crop evapotranspiration, expressed as evapotranspiration). The values are presented as the mean ± standard deviation for three replicates. Small bracketed alphabets (a-s) denote statistical significance at a 95% confidence interval.

Relative water, proline and sugar content of maize leaves

The results of relative water content, proline and sugar content at two growth stages is presented in Table 2. At the V14th stage, DK-9108 plants in 4-tonnes acre−1 AB amended soil under 70–50% IW demonstrated the highest relative water content (34.8–55%) compared to those planted without AB amended soil. At the R3 stage, DK-6321 had a significantly higher relative water content (8–10%) in 4-tonnes-acre−1 AB amended soil than in DK-6321 grown in non-amended soil under 70–50% IW. While DK-9108 at the V14th stage without AB amendment in soil under 50% IW showed a notable reduction in relative water content. When compared to Sarhaab produced under 100% IW, Sarhaab under 50% IW had the lowest relative water content (38.8%) at the R3 stage in soil without AB amendment.

Table 2 The impact of organically activated biochar on relative water content, proline content, and sugar content at V14 (fourteen leaf stage) and R3 (milking) stages of maize under different moisture conditions.

Treatment	V14	R3	V14	R3	V14	R3	
	Relative Water Content	Proline Content	Sugar Content	
V 1 ×T 0 ×100%IW	80.86 ± 1.62 (e-h)	61.43 ± 0.67 (c-e)	1.646 ± 0.11 (e)	6.192 ± 0.02 (e-f)	0.342 ± 0.01 (h)	0.4986 ± 0.005 (d)	
V 1 ×T 0 ×70%IW	72.14 ± 1.44 (i-k)	52.64 ± 1.67 (i)	2.139 ± 0.04 (c)	6.397 ± 0.01 (d)	0.355 ± 0.003 (f-g)	0.5006 ± 0.005 (c-d)	
V 1 ×T 0 ×50%IW	48.73 ± 2.2 (l)	37.71 ± 1.29 (l-m)	2.589 ± 0.02 (a)	6.685 ± 0.02 (b)	0.376 ± 0.001 (c-d)	0.5078 ± 0.001 (b)	
V 1 ×T 1 ×100%IW	90.03 ± 2.9 (c)	64.02 ± 0.96 (b-c)	0.640 ± 0.02 (m-n)	4.221 ± 0.01 (p)	0.271 ± 0.002 (k)	0.4856 ± 0.005 (h-i)	
V 1 ×T 1 ×70%IW	86.75 ± 3.31 (c-e)	53.65 ± 1.25 (g-i)	0.591 ± 0.02 (n)	4.563 ± 0.02 (o)	0.299 ± 0.001 (j)	0.4876 ± 0.005 (g-i)	
V 1 ×T 1 ×50%IW	68.72 ± 1.75 (j-k)	40.13 ± 1.28 (k-l)	1.084 ± 0.03 (h-i)	4.894 ± 0.02 (m)	0.318 ± 0.002 (i)	0.4892 ± 0.004 (f-h)	
V 1 ×T 2 ×100%IW	113.05 ± 1.34 (a)	65.71 ± 0.26 (b)	0.745 ± 0.02 (l-m)	5.092 ± 0.02 (l)	0.242 ± 0.001 (m)	0.4746 ± 0.005 (m)	
V 1 ×T 2 ×70%IW	88.87 ± 1.60 (c-e)	54.07 ± 1.56 (g-i)	0.332 ± 0.01 (o)	5.435 ± 0.02 (j)	0.258 ± 0.001 (l)	0.4763 ± 0.005 (l-m)	
V 1 ×T 2 ×50%IW	75.71 ± 1.53 (g-h)	43.37 ± 0.58 (j-k)	1.319 ± 0.05 (f)	5.808 ± 0.03 (h)	0.298 ± 0.003 (j)	0.4774 ± 0.005 (k-m)	
V 2 ×T 0 ×100%IW	77.38 ± 1.43 (f-i)	62.78 ± 0.52 (b-d)	1.701 ± 0.02 (e)	5.874 ± 0.01 (h)	0.363 ± 0.002 (e)	0.4976 ± 0.002 (d-e)	
V 2 ×T 0 ×70%IW	65.92 ± 3.51 (k)	52.80 ± 1.32 (i)	1.948 ± 0.02 (d)	6.168 ± 0.03 (f)	0.375 ± 0.003 (c-d)	0.5008 ± 0.005 (c-d)	
V 2 ×T 0 ×50%IW	54.24 ± 3.22 (l)	40.83 ± 1.03 (k-l)	2.515 ± 0.02 (a-b)	6.433 ± 0.01 (d)	0.405 ± 0.001 (b)	0.5037 ± 0.005 (b-c)	
V 2 ×T 1 ×100%IW	85.00 ± 3.19 (c-f)	69.91 ± 1.69 (a)	0.795 ± 0.02 (k-l)	3.722 ± 0.01 (q)	0.351 ± 0.01 (g)	0.4896 ± 0.001 (f-h)	
V 2 ×T 1 ×70%IW	82.46 ± 1.49 (c-h)	57.49 ± 2.55 (e-g)	0.616 ± 0.02 (n)	4.197 ± 0.02 (p)	0.361 ± 0.003 (e-f)	0.4913 ± 0.001 (f-g)	
V 2 ×T 1 ×50%IW	64.43 ± 1.63 (k)	42.45 ± 0.59 (j-k)	0.751 ± 0.01 (l-m)	4.551 ± 0.02 (o)	0.381 ± 0.001 (c-d)	0.4933 ± 0.002 (e-f)	
V 2 ×T 2 ×100%IW	105.91 ± 1.30 (a-b)	72.53 ± 1.57 (a)	1.121 ± 0.06 (g-h)	4.774 ± 0.02 (n)	0.302 ± 0.002 (j)	0.4809 ± 0.005 (j-k)	
V 2 ×T 2 ×70%IW	89.21 ± 1.46 (c-d)	57.07 ± 2 (f-h)	1.047 ± 0.03 (h-i)	5.122 ± 0.02 (l)	0.316 ± 0.002 (i)	0.4787 ± 0.005 (k-m)	
V 2 ×T 2 ×50%IW	75.48 ± 1.78 (g-j)	43.09 ± 0.61 (j-k)	1.084 ± 0.01 (h-i)	5.399 ± 0.01 (j)	0.342 ± 0.002 (h)	0.4791 ± 0.005 (k-l)	
V 3 ×T 0 ×100%IW	74.47 ± 0.57 (h-j)	58.98 ± 0.49 (d-f)	2.441 ± 0.02 (b)	6.240 ± 0.02 (e)	0.373 ± 0.002 (d)	0.5275 ± 0.001 (a)	
V 3 ×T 0 ×70%IW	64.24 ± 1.04 (k)	46.14 ± 1.35 (j)	2.416 ± 0.03 (b)	6.595 ± 0.01 (c)	0.383 ± 0.001 (c)	0.5258 ± 0.001 (a)	
V 3 ×T 0 ×50%IW	53.79 ± 2.61 (l)	36.07 ± 0.63 (m)	2.626 ± 0.02 (a)	6.902 ± 0.02 (a)	0.421 ± 0.002 (a)	0.5285 ± 0.001 (a)	
V 3 ×T 1 ×100%IW	82.93 ± 2 (c-g)	62.87 ± 1.32 (b-d)	0.980 ± 0.01 (i-j)	5.291 ± 0.02 (k)	0.364 ± 0.01 (e)	0.5031 ± 0.005 (c)	
V 3 ×T 1 ×70%IW	81.79 ± 2.93 (d-h)	53.33 ± 0.99 (h-i)	0.912 ± 0.05 (j-k)	5.645 ± 0.02 (i)	0.379 ± 0.003 (c-d)	0.5038 ± 0.001 (b-c)	
V 3 ×T 1 ×50%IW	69.59 ± 0.91 (i-k)	41.11 ± 1.11 (k-l)	1.072 ± 0.01 (h-i)	5.964 ± 0.01 (g)	0.407 ± 0.001 (b)	0.5048 ± 0.001 (b-c)	
V 3 ×T 2 ×100%IW	101.22 ± 1.36 (b)	62.84 ± 1.01 (b-d)	1.152 ± 0.02 (g-h)	6.144 ± 0.01 (f)	0.321 ± 0.002 (i)	0.4842 ± 0.002 (i-j)	
V 3 ×T 2 ×70%IW	82.68 ± 1.37 (c-g)	51.63 ± 1.62 (i)	1.232 ± 0.02 (f-g)	6.379 ± 0.02 (d)	0.337 ± 0.001 (h)	0.4791 ± 0.001 (k-l)	
V 3 ×T 2 ×50%IW	70.20 ± 1.68 (i-k)	43.56 ± 1.23 (j-k)	1.276 ± 0.02 (f)	6.649 ± 0.01 (b-c)	0.375 ± 0.002 (d)	0.4843 ± 0.001 (k-l)	
Notes.

The results represent the means derived from a three-way analysis of variance (ANOVA) involving maize varieties (V1= DK-9108, V22= DK-6321, and V3= Sarhaab) activated biochar amendment levels (T0= 0 tonnes acre−1, T1= 2 tonnes acre−1, T2= 4 tonnes acre−1), and irrigation water levels (100% IW= 100% irrigation water of crop evapotranspiration, expressed as evapotranspiration, 70% IW= 70% irrigation water of crop evapotranspiration, expressed as evapotranspiration, 50% IW= 100% irrigation water of crop evapotranspiration, expressed as evapotranspiration). The values are presented as the mean ± standard deviation for three replicates. Small bracketed alphabets (a-s) denote statistical significance at a 95% confidence interval.

All maize hybrids cultivated in soil without AB amendment exhibited significantly elevated proline content under 50% IW. However, Sarhaab had the highest proline content at V14th stage (7.49%) and R3 stage (8.93%) without AB amendment in soil under 50% IW, compared to Sarhaab grown in 100% IW. DK-9108 showed a considerable drop in proline content (96–104%) during the V14th stage in 4-tonnes acre−1 AB amended soil under 70%–50% IW, compared to DK-9108 produced without AB amendment. At the R3 stage, DK-6321 had significantly lower proline content (19–20%) in 4-tonnes acre−1 AB amended soil under 100% IW compared to plants of soil without AB amendment under 50%–70% IW.

At V14th stage (12%) and R3 stage (6%) the highest sugar content was observed in Sarhaab without AB amendment in soil under 50% IW in contrast with 100% IW. However, lowest sugar content was found in DK-6321 (−6–18% and −10.7–22%) at the V14th stage in 2 and 4-tonnes acre−1 AB amended soil under 50% and 70% IW, respectively, in contrast to plants grown without AB amendment in soil. At the R3 stage, sugar content slightly declined in all maize hybrids (2.1–2.5% and 3.9–5%) grown in 2 and 4-tonnes acre−1 AB amended soil in contrast to plants grown without AB amendment in soil under 50% and 70% IW.

Leaf FTIR

The leaf FTIR spectra of three verities are presented in Fig. 2. The peak at 3700–3000 cm−1 signifies the stretching of phenolics’ OH groups. When compared to 100% IW, the peak intensity of phenolics in DK-9108 decreased by 7%–9% at 70% and 50% IW in soil without AB amendment. However, with a 2 tonnes acre−1 AB amendment in soil, the peak intensity of phenolics was slightly increased by 1.5% and 2.4% under 70% and 50% IW than 100% IW. In DK-6321, the peak intensity of phenolics increased by 2.5–7% in 4 tonnes acre−1 AB amended soil under 70%–50% IW, as compared to plants grown in soil without AB amendment. The minor peak at 2,900 cm−1 is caused by stretching vibration of C =O from alkanes, and its lowest intensity was seen in DK-9108 under 100% IW without AB amendment in the soil. The intensity of alkanes increased by 17–80% in DK-6321 plants grown in soil amended with 4 tonnes acre−1 AB under 70%–50% IW. The peaks at 1,700 cm−1 show the stretching vibration of C =O from alkyl esters and pectin. The peak strength increased by 66% and 93% in DK-6321 plants in 2 and 4 tonnes acre−1 of AB amended soil under 70% IW. The 1,644 cm−1, 1,384 cm−1, and 1,247 cm−1 peaks show the C-N stretching vibration of amide I, amide II, and amide III, respectively. A decrease in the level of irrigation water in DK-9108, DK-6321, and Sarhaab results in an increase in the peak intensities of amides. The peak at 1,000–1,100 cm−1 represents the stretching vibration of the C-O, C-H, and C − O − C from carbohydrates. Compared to plants cultivated in soil without an AB amendment, the peak intensity of carbohydrates increased under 50% IW in 2 and 4 tonnes acre−1 AB amended soil, with plants of 4 tonnes acre−1 showing a 43% greater intensity. In DK-6321, under 100%, 70%, and 50% of IW, the peak intensity of carbohydrates rose in both levels of AB amended soil.

Figure 2 The impact of organically activated biochar (T0; 0 tonnes acre−1, T1; 2 tonnes acre−1, T2; 4 tonnes acre−1) on secondary metabolites of maize leaves under 100%, 70% and 50% IW (irrigation water of crop evapotranspiration) (A) V1 = DK-9108 (B) V2 = DK-6321 (C) V3 = Sarhaab.

Morphological growth attributes

The results pertaining to maize plant morphological attributes, derived from a three-way analysis of variance (ANOVA), are detailed in Table 3. The maize plant in AB amended soil exhibited longer roots and shoots. The DK-6321 cultivated in a 4-tonnes acre−1 AB amendment level showed enhanced root length (48.85%–54.14%) and shoot length (10.21%–99%) under 50%–70% IW compared to DK-6321 grown in soil without AB amendment. In soil without AB amendment, when compared to Sarhaab grown under 100% IW, a significant reduction in root length (5.27%) and shoot length (2%) was seen under 50% IW. The DK-6321 plants exhibited significantly higher root fresh weight (39.40%–43.07%) and shoot fresh weight (5%–6.5%) in soil treated with 4-tonnes acre−1 of AB amendment under 50%–70% IW, compared to the DK-6321 grown in soil without AB amendment under 50%–70% IW. When compared to Sarhaab grown in soil without AB amendment under 100% IW, shoot and root fresh weight decreased significantly (1.5%–5.9%) under 50% IW. In 4-tonnes acre−1 AB amended soil 45.54%–48.87% increased leaf fresh weight was recorded in DK-6321 under 70%–50% IW when compared with DK-6321 in soil without AB amendment under 70%–50% IW. The lowest leaf fresh weight was recorded in Sarhaab (10.2%) grown in soil without AB amendment under 50% IW, in contrast to Sarhaab under 100% IW. In DK-6321 plants cultivated in 4-tonnes acre−1 AB amended soil with 50–70% IW had a higher root dry weight (45.5%–84.9%) and shoot dry weight (26.04%–25%) compared to DK-6321 in soil without AB amendment under the same IW. The root dry weight and shoot dry weight of Sarhaab cultivated without AB amendment in soil under 50% IW decreased by 20.75% and 79%, respectively, compared to 100% IW.

Table 3 The impact of organically activated biochar on root fresh weight, shoot fresh weight, leaf fresh weight, root dry weight, shoot dry weight, shoot length, root length of maize under different moisture conditions.

Treatment	Root fresh weight (g)	Shoot fresh weight (g)	Leaf fresh weight (g)	Root dry weight (g)	Shoot dry weight (g)	Leaf dry weight (g)	Soot length (cm)	Root length (cm)	
V 1 ×T 0 ×100%IW	168.42 ± 2.33 (n-o)	1107.45 ± 0.01 (q-r)	162.02 ± 1.67 (o)	75.15 ± (n-o)	240.45 ± 0.01 (q-r)	59.02 ± 0.01 (o)	252.71 ± 0.63 (m-n)	62.65 ± 1.38 (l-m)	
V 1 ×T 0 ×70%IW	174.30 ± 0.89 (m-n)	1114.82 ± 0.01 (o-p)	156.82 ± 1.49 (o)	81.03 ± (m-n)	247.82 ± 0.01 (o-p)	53.82 ± 0.01 (o)	258.90 ± 1.09 (k-l)	67.35 ± 1.72 (k-l)	
V 1 ×T 0 ×50%IW	163.18 ± 1.41 (o)	1100.27 ± 0.01 (s-t)	150.68 ± 1.28 (p)	69.91 ± (o)	233.27 ± 0.01 (s-t)	47.68 ± 0.01 (p)	245.60 ± 1.81 (o-p)	57.01 ± 1.71 (n-o)	
V 1 ×T 1 ×100%IW	191.78 ± 1.53 (j-k)	1140.85 ± 0.01 (i)	199.70 ± 1.68 (i)	98.51 ± (j-k)	273.85 ± 0.29 (i)	96.70 ± 0.01 (i)	275.49 ± 1.60 (d-g)	79.12 ± 1.33 (h-i)	
V 1 ×T 1 ×70%IW	197.75 ± 1.92 (i-j)	1147.05 ± 0.29 (h)	195.55 ± 1.89 (i-j)	104.48 ± (i-j)	280.05 ± 0.01 (h)	92.55 ± 0.01(i-j)	279.77 ± 1.74 (b-e)	84.85 ± 1.78 (e-g)	
V 1 ×T 1 ×50%IW	186.62 ± 1.85 (k-l)	1137.46 ± 0.01 (i-j)	187.46 ± 2.5 (k)	93.35 ± (k-m)	270.46 ± 0.01 (i-j)	84.46 ± 0.01 (k)	271.97 ± 1.47 (f-g)	74.30 ± 4.04 (j)	
V 1 ×T 2 ×100%IW	231.11 ± 1.38 (e)	1179.06 ± 0.02 (d)	228.02 ± 1.14 (f)	137.84 ± (e)	312.06 ± 0.01 (d)	125.02 ± 0.01 (f)	277.31 ± 0.15 (c-f)	93.85 ± 0.86 (b-c)	
V 1 ×T 2 ×70%IW	239.32 ± 1.18 (d)	1184.96 ± 0.01 (c)	218.41 ± 1.76 (g)	146.05 ± (d)	317.96 ± 0.01 (c)	115.41 ± 0.01(g)	281.95 ± 1.9 (b-c)	97.79 ± 1.72 (b)	
V 1 ×T 2 ×50%IW	224.57 ± 1.31 (e-f)	1171.75 ± 0.01 (e)	209.32 ± 2.24 (h)	131.30 ± (e-f)	304.75 ± 0.01 (e)	106.32 ± 0.01 (h)	271.45 ± 1.63 (g-h)	87.94 ± 0.95 (d-f)	
V 2 ×T 0 ×100%IW	183.95 ± 1.18 (l)	1130.47 ± 0.01 (k-l)	183.8 ± 1.53 (k)	90.68 ± (l)	263.47 ± 0.01 (k-l)	80.8 ± 0.01 (k)	260.38 ± 1.75 (j-l)	67.16 ± 0.91 (k-l)	
V 2 ×T 0 ×70%IW	189.37 ± 1.36 (k-l)	1136.45 ± 0.01 (i-j)	176.65 ± 1.34 (l)	96.10 ± (k-l)	269.45 ± 0.01 (i-j)	73.65 ± 0.01 (l)	262.75 ± 1.76 (i-k)	71.83 ± 1.75 (j-k)	
V 2 ×T 0 ×50%IW	176.89 ± 1.48 (m)	1122.82 ± 0.01 (m-n)	168.59 ± 1.68 (n)	83.62 ± (m)	255.82 ± 0.01 (m-o)	65.59 ± 0.01 (n)	255.41 ± 1.21 (l-m)	61.34 ± 1.18 (m-n)	
V 2 ×T 1 ×100%IW	212.46 ± 2.42 (g)	1155.70 ± 0.01 (g)	243.13 ± 1.18 (d)	119.19 ± (g)	288.7 ± 0.01 (g)	140.13 ± 0.01 (d-f)	283.08 ± 1.15 (b)	84.63 ± 0.16 (f-g)	
V 2 ×T 1 ×70%IW	228.31 ± 1.23 (e)	1162.06 ± 0.01 (f)	235.47 ± 2.34 (e)	135.04 ± (e)	295.06 ± 0.01 (f)	132.47 ± 0.01 (e)	280.43 ± 1.74 (b-d)	88.87 ± 0.57 (d-f)	
V 2 ×T 1 ×50%IW	203.35 ± 2.09 (h-i)	1140.12 ± 0.01 (i)	225.61 ± 1.57 (f)	110.08 ± (h-j)	273.12 ± 0.76 (i)	122.61 ± 0.01 (f)	274.19 ± 0.55 (e-g)	71.43 ± 0.80 (j-k)	
V 2 ×T 2 ×100%IW	258.51 ± 1.89 (b)	1195.91 ± 0.01 (b)	264.89 ± 1.5 (a)	165.24 ± (b)	328.91 ± 0.01 (b)	161.89 ± 0.01 (a)	284.89 ± 1.97 (a-b)	96.26 ± 1.01 (b)	
V 2 ×T 2 ×70%IW	270.93 ± 2.1 (a)	1203.75 ± 0.01 (a)	257.11 ± 1.84 (b)	177.66 ± (a)	336.75 ± 0.01 (a)	154.11 ± 0.01 (b)	289.57 ± 1.28 (a)	106.91 ± 1.61 (a)	
V 2 ×T 2 ×50%IW	246.59 ± 1.58 (c)	1189.44 ± 0.77 (c)	250.98 ± 1.5 (c)	153.32 ± (c)	322.44 ± 0.01 (c)	147.98 ± 0.03 (c)	280.59 ± 1.75 (b-d)	90 ± 0.59 (c-d)	
V 3 ×T 0 ×100%IW	130.55 ± 1.99 (p)	1103.33 ± 0.01 (r)	146.89 ± 1.5 (p)	37.28 ± (p-q)	236.33 ± 0.01 (r)	43.89 ± 0.01 (p)	244.89 ± 1.97 (o-p)	55.51 ± 0.89 (o)	
V 3 ×T 0 ×70%IW	136.55 ± 1.27 (p)	1110.92 ± 0.01 (p-q)	139.06 ± 1.22 (q)	43.28 ± (p)	243.92 ± 0.01 (p-r)	36.06 ± 0.01 (q)	249.29 ± 1.37 (d-g)	60.58 ± 0.87 (m-n)	
V 3 ×T 0 ×50%IW	122.82 ± 1.48 (q)	1095.43 ± 0.01 (s)	131.89 ± 1.7 (m-n)	29.55 ± (q)	228.43 ± 0.01 (s)	28.89 ± 0.01 (m-n)	275.19 ± 1.62 (n-o)	52.58 ± 0.45 (o)	
V 3 ×T 1 ×100%IW	167.65 ± 0.79 (n-o)	1120.26 ± 0.01 (n)	175.15 ± 1.15 (r)	74.38 ± (n-o)	253.26 ± 0.01 (n)	72.15 ± 0.01 (r)	240.54 ± 1.25 (p)	75.30 ± 0.46 (i-j)	
V 3 ×T 1 ×70%IW	173.32 ± 1.84 (m-n)	1125.65 ± 0.01 (l-m)	170.48 ± 1.49 (l-m)	80.05 ± (m-n)	258.65 ± 0.01 (l-m)	67.48 ± 0.01 (l-m)	272.46 ± 1.18 (f-h)	80.37 ± 1.19 (g-h)	
V 3 ×T 1 ×50%IW	162.41 ± 1.13 (o)	1117.97 ± 0.01 (n-o)	167.51 ± 2.29 (n)	69.14 ± (o)	250.97 ± 0.01 (n-o)	64.51 ± 0.01 (o)	265.66 ± 1.43 (i-j)	71.23 ± 0.66 (j-k)	
V 3 ×T 2 ×100%IW	213.41 ± 1.73 (g)	1126.31 ± 0.01 (l-m)	193.25 ± 2.05 (j)	120.14 ± (g)	259.31 ± 0.01 (l-m)	90.25 ± 0.05 (g)	276.62 ± 1.30 (c-g)	84.71 ± 0.62 (f-g)	
V 3 ×T 2 ×70%IW	220.40 ± 1.75 (f)	1132.91 ± 0.01 (j-k)	184.07 ± 1.27 (k)	127.13 ± (f)	265.91 ± 0.01 (j-k)	81.07 ± 0.01 (f)	280.58 ± 1.44 (b-d)	89.54 ± 1.62 (c-e)	
V 3 ×T 2 ×50%IW	206.61 ± 2.81 (g-h)	1122.40 ± 0.01 (m-n)	178.36 ± 1.25 (l)	113.34 ± (g-h)	255.40 ± 0.01 (m-n)	75.36 ± 0.01 (g-h)	267.52 ± 1.75 (h-i)	79.17 ± 0.52 (h-i)	
Notes.

The results represent the means derived from a three-way analysis of variance (ANOVA) involving maize varieties (V1= DK-9108, V22= DK-6321, and V3= Sarhaab) activated biochar amendment levels (T0= 0 tonnes acre−1, T1= 2 tonnes acre−1, T2= 4 tonnes acre−1), and irrigation water levels (100% IW= 100% irrigation water of crop evapotranspiration, expressed as evapotranspiration, 70% IW= 70% irrigation water of crop evapotranspiration, expressed as evapotranspiration, 50% IW= 100% irrigation water of crop evapotranspiration, expressed as evapotranspiration). The values are presented as the mean ± standard deviation for three replicates. Small bracketed alphabets (a-s) denote statistical significance at a 95% confidence interval.

Yield attributes

The results of the three-way analysis of variance (ANOVA) for maize plant yield characteristics are delineated in Fig. 3. All maize hybrids underwent a reduction in ear length due to water stress; DK–9108 under 50% IW and soil without AB amendment had the lowest ear length (54.14%) compared to DK-9108 under 100% IW. The DK-6321 had significantly longer ear length (9.89%–29.48%) when grown in 4-tonnes acre−1 AB amended soil under 50%–70% IW compared to plants cultivated in soil without AB amendment. At 70%–50% IW, DK-6321 showed increased ear weight (164.9%–172.8%) in 4-tonnes acre−1 AB amended soil compared to DK-6321 grown in soil without AB amended soil. A significant decline of 8.62% was observed in soil without AB amendment in DK-9108 under 50% IW when compared with DK-9108 under 100% IW. The total number of seeds per ear was recorded to be highest (31.29%) in DK-6321 under 70% IW in 4-tonnes acre−1 AB amended soil by 31.29% as compared with DK-6321 under 70% IW in non-amended soil. The result showed a 157%–162.64% increase in 1,000-grain weight in DK-6321 under 70%–50% IW in soil amended with 4-tonnes acre−1 when compared with DK-6321 in soil without AB amendment. In contrast to Sarhaab grown in soil without AB amendment under 100% IW, there was a 26.5% decrease in 1,000-grain weight under 50% IW.

Figure 3 The impact of organically activated biochar (T0; 0 tonnes acre−1, T1; 2 tonnes acre−1, T2; 4 tonnes acre−1) on ear length, ear weight, total number of grains/ears, 1,000 grain weight of maize under 100%, 70% and 50% IW (irrigation water of crop evapotranspiration).

Multivariate analysis

The heatmap (Fig. 4) illustrates the response of three maize varieties, DK-6321, DK-9108, and Sarhaab, to different levels of activated biochar (0-tonnes acre−1 AB, 2-tonnes acre−1 AB, and 4-tonnes acre−1 AB) under varying irrigation regimes (100%, 70%, and 50% of ETc). Each row represents a combination of specific maize varieties, biochar amendments, and irrigation levels, while each column represents the physio-biochemical and yield attributes of maize. The color intensity within each cell indicates the performance of the respective maize variety under the specified treatment conditions, with darker red shades representing higher performance and darker blue shades representing lower performance. The highest improvement in photosynthetic pigments, relative water content, morphological parameters, and yield parameters is shown by DK-6321 under 100%, 70%, and 50% irrigation of ETc, with 4-tonnes acre−1 levels of activated biochar, as evidenced by the darker shading in the corresponding cells. Conversely, Sarhaab showed optimal performance at lower levels of activated biochar under 50% irrigation of ETc, as indicated by the light blue shading in the respective cells.

Figure 4 The impact of organically activated biochar (T0; 0 tonnes acre−1, T1; 2 tonnes acre−1, T2; 4 tonnes acre−1) on Caro V14 (carotenoid content at V14 stage), Chl b V14 (chlorophyll b at V14 stage), T Chl R3 (total chlorophyl content at R3; milking stage), Chl a R3 (chlorophyll a at R3 stage), T Chl V14 (total chlorophyll content at V14 stage), T Chl V14 (total chlorophyll content at V14 stage), Chl a V14 (chlorophyll a at V14 stage), SDW (shoot dry weight), SFW (shoot fresh weight), RDW (root dry weight), RFW (root fresh weight), LDW (leaf dry weight), LFW (leaf fresh weight), TGW (total grain weight), EW (ear weight), SL (shoot length), TNOG (total number of grains), RL (root length), EL (ear length), Caro R3 (carotenoid content at R3 stage), Chl b R3 (Chlorophyll b at content R3 stage), RWC R3 (relative water content at R3 stage), RWC at V14 (relative water content at V14 stage), SC R3 (sugar content at R3 stage), SC V14 (sugar content at V14 stage), Pro R3 (proline content at R3), Pro V14 (proline content at V14 stage) of maize plants (V1 = DK-9108, V2 = DK-6321, and Sarhaab) under 100%, 70% and 50% irrigation water (IW) of crop evapotranspiration.

Discussion

Physicochemical properties of soil

The AB amendment improved the soil organic carbon and organic matter content but also caused a slight increase in electrical conductivity that might be due to the higher nutrient and organic carbon content of biochar (Joseph et al., 2020; Qi et al., 2020). Moreover, AB can stimulate the polymerization of small organic particles via a surface catalytic reaction; therefore, macropores of biochar may absorb small organic molecules from the soil (Zhang, Wang & Feng, 2021). In the current study, it was observed that the pH of AB amended soil was slightly acidic. This decline in soil pH was most likely caused by the oxidation of biochar and production of acids, facilitated by organic matter in the soil (Premalatha et al., 2023). Simultaneously, biochar also caused the oxidation of organic matter, resulting in the formation of humic acid and other organic acids (Mittal et al., 2023). This process may have caused a minor pH reduction in soil (Huang et al., 2019a). At present, it was also observed that the AB amended soil had the highest potassium and phosphorus content in contrast to the non-amended soil. This can be due to the highest potassium and phosphorus content of wood biochar, which becomes readily available to plants after dissolving in soil water (Jindo et al., 2020). Increased K absorption is crucial for maintaining guard cell pressure, which controls stomatal conductance, and for achieving optimal osmoregulation (Brantley et al., 2016). In order to improve soil fertility, the organic content of vermicompost enhances the process of nutrient cycling and mineralization controlled by the soil microbial population (Igalavithana et al., 2017). The AB amendment enhances soil characteristics such as enhanced WHC, slightly acidic pH, reduced nutrient leaching, delivers critical nutrients to the soil on its own, and protects crops from water stress (Yang et al., 2018; Fahad et al., 2022). In fact, activated biochar is a helpful organic soil supplement that raises the soil’s fertility. Both chemical and organic activation of biochar can improve the surface area and water retention capacity of soil, but organic activation of biochar is preferred due to its safer impact on the agroecological system (Ghassemi-Golezani & Farhangi-Abriz, 2022; Jahan et al., 2023). Mohamed et al. (2016) reported that 2% chemically engineered biochar caused an increase in water holding capacity of loamy sand soil by 30% after an incubation period of six weeks, while Jahan et al. (2023) reported an increase in water holding capacity of 31% to 41% in 2.5% and 5% organically activated biochar amended loamy sand soil after a period of 20 weeks.

Physiological and biochemical attributes

To cope with drought, plants evolved a variety of morphological, biochemical, and physiological adaptations, as changes in shoot-to-root ratio, leaf area, accumulation of various osmolytes (proline and soluble sugars), membrane stability, photosynthetic pigments, and relative water content (Farooq et al., 2024; Shirvani et al., 2024). By holding the right amount of moisture in its tiny micropores and making it available to plants during the drought, biochar supplementation enhanced the growth qualities of crops under water scarce conditions (Le Guyader et al., 2024).

The chlorophyll a and b content and total chlorophyll and carotenoid content were significantly reduced at 50% IW in all maize hybrids, and the highest decline was noted in Sarhaab at 50% IW (indicating the sensitivity of Sarhaab to water stress). This reduction might be due to overproduction of the reactive oxidative species (Jahan et al., 2023). These ROS triggered photo-oxidation and, hence, degradation of the ultrastructure of the thylakoid membrane and photosystems, reducing the swelling of stromal lamella, which ultimately destroys the thylakoid and antenna complexes (Huang et al., 2019). These alterations were responsible for the overall decline in photosynthetic pigments due to the loss of PS II and light harvesting complexes (Rehman et al., 2023). However, with the addition of AB amendment to the soil, an improvement in chlorophyll a and b content in the maize hybrids (DK-9108, DK0-6321, and Sarhaab) was detected even at 50% IW. The highest chlorophyll a and b content was recorded in DK-9108 at both the V14th and R3 stages, which might be an indication of the harmony of this maize hybrid with the AB amendment in the soil. The diverse pore range of the AB has the ability to retain moisture for a longer duration and make it available for plants under water-scarce conditions (Ansari et al., 2020). So, this gradual supply of moisture to the maize plant prevents the thylakoid membrane from shrinking and improves photosynthetic pigments, particularly in DK-9108, even under drought stress (Ahmad et al., 2021; Jafari, Garmdareh & Azadegan, 2019; Tayyab et al., 2018).

The proline content was enhanced as the severity of water stress increased in the leaves of all maize hybrids grown in soil without AB amendment, and the highest proline content was detected in Sarhaab. Proline is a stress indicator and acts like a compatible osmolyte, which tends to maintain the internal cellular organisation under stress conditions. It is also very important for keeping plant cell membranes healthy by getting rid of extra reactive oxygen species and keeping plant cell redox reactions safe (Ahmed et al., 2018). However, in 2-4-tonnes acre−1 AB amended soil at both V14th and R3 growth stages, low proline content in the leaves of the maize hybrid was noted even under 50%–70% IW. This reduction in proline content, along with an enhancement in photosynthetic pigments, further justified the adequate moisture provision potential of biochar to plant roots even under deficit irrigation, which consequently helps the maize plants maintain normal redox balance even under water stress conditions (Mannan et al., 2017).

When maize plants were cultivated without an AB amendment in the soil during the V14th and R3 stages, Sarhaab showed the maximum sugar content under 50% IW. Similarly, DK-9108 and DK-6321 also showed increased sugar content. This could be because the breakdown of carbohydrates stabilises the proper cellular osmotic potential, maintaining optimal water uptake into cells to achieve maximum turgidity (Du et al., 2020; Thalmann & Santelia, 2017). On the contrary, plants cultivated in soil supplemented with biochar exhibited a low sugar content. This might be the result of AB facilitating increased moisture availability (Yadav et al., 2018). The low sugar accumulation in all maize hybrids at both AB amendment levels (2-tonnes acre−1 and 4-tonnes acre−1), particularly under 70% and 50% IW in AB amended soil, is evidence that the plants were able to withstand water stress conditions.

All three maize hybrids had lower relative water content because, under water stress, plants closed their stomata to reduce transpiration. However, it also influences water conductance from roots to leaves, lowering leaf water content (Luo et al., 2020). Relative water content is an essential measure of plant water status (Bistgani et al., 2017). Maize hybrids grown on AB-amended soil (2-4 acres) showed higher relative water content in the leaves of DK-9108, DK-6321, and Sarhaab. The AB integrated recalcitrant carbon content with high porosity, which improves soil’s water retention efficacy by releasing moisture to roots and improving relative water content in plants under water stress circumstances (Jahan et al., 2022; Ullah et al., 2020).

Drought and plant metabolites

The FTIR analysis of plants is the most convenient approach for the identification of different inorganic and organic compounds present in them. After screening, phenolic, alkane, alkyl-esters, pectin, amides (I-II-III), proteins, and carbohydrates functional groups were found. Phenolic compounds usually contain OH groups and are produced in plants during water stress conditions. All maize plants grown in soil without AB amendment showed the highest peak intensity of phenolics under water stress (70%–50% IW). Basically, water stress stimulates ROS production in plants; however, to protect themselves from oxidative damage, plants synthesise various secondary metabolites, and phenolics are one of them (Stiller et al., 2021; Baliyan et al., 2022). Phenolics are non-enzymatic antioxidants that are associated with scavenging free radicals (Hasanuzzaman et al., 2021; Zulfiqar et al., 2023). As the intensity of water stress increases, the biosynthesis of phenolic compounds also enhances (Parveen et al., 2019). While the phenolic peaks were detected to be less intense in DK-9108, DK-6321, and Sarhaab in soil with 2-4 tonne acre−1 AB amendment levels, even at 50% and 70%, this might be due to the appropriate moisture and nutrient provision to plants by highly porous, spongy biochar. The moisture and nutrients are supplied to the plants gradually to withstand water stress intervals. So, maize hybrids cultivated under AB amended soil reduced the phenolic content due to less production of ROS (Gharred et al., 2022).

The band intensities at 2,900 cm−1, 1,850–1,250 cm−1, and 1,000–900 cm−1 suggest alkanes, alkyl ester, amides, polysaccharides, and proteins (Lalay et al., 2024). Currently, all hybrid maize varieties show a decrease in peak intensities at 70%–50% IW in soils without AB amendment. A reduction in peak intensities of amides, proteins and polysaccharides under stress was also reported by Ogbaga et al. (2017). Water stress destroyed protein content and slowed down the production of chlorophyll. This led to a huge drop in photosynthesis, which in turn decreased the leaf’s carbohydrate content (Mahmoud et al., 2022). While higher peak intensities were observed in the DK-9108, DK-6321, and Sarhaab grown in AB amended soil under all IW levels. The alkane and alkyl ester content was increased in the plants grown in AB amended soil, which makes them more resistant to water stress by synthesizing wax layers. Increased alkane chain lengths of C29 and C31 improved drought resistance (Li et al., 2020). Water permeability from cuticular wax is not dependent on its overall covering, but rather on its composition, particularly on the nonpolar component (mostly alkanes), which functions as a very effective water barrier (Tunstad et al., 2024). Pectin is the primary component of the cell wall and aids in cell growth. Plant cell walls protect against drought stress because of a trait that is dependent on several of its fundamental constituents, one of which is pectin. Our FTIR data revealed an increase in pectin concentration in maize plant leaves grown in AB amended soil, making them more resistant to water stress and aiding in the maintenance of optimal cell size and shape for normal cell communication and function (Forand et al., 2022; Shin et al., 2021). The increased protein and carbohydrate content of plants grown in AB-amended soil demonstrated that AB, by offering the highest possible level of hydration, mitigated the negative effects of water stress on carbon absorption and photosynthetic capacity (Wang et al., 2021). The AB provides plants with an abundance of carbon and nitrogen, which facilitates the production of chlorophyll, sucrose, and protein. As a result, the leaves ultimately exhibited an increase in carbohydrate and amide concentrations (Gharred et al., 2022; Khan et al., 2021; Singh et al., 2019).

Morphological and yield attributes of plants

Water stress significantly reduced the fresh and dry weights of root, shoot, and leaf in all maize hybrids (DK-9108, DK-6321, and Sarhaab), with Sarhaab showing the greatest loss at 50% IW in soil. The possible cause for this phenomenon could be a reduction in cell turgidity resulting from a diminished water potential across maize plant cells, which subsequently leads to the contraction of cell walls and impedes cell division (Carvalho et al., 2021; Shakeel et al., 2022). Moreover, the collapse of organelles (chloroplasts, mitochondria) in flaccid cells resulted in a decrease in photosynthesis, which eventually lowered plant biomass (Endres et al., 2019; Keipp et al., 2020). Water stress induces a reduction in plant growth via a variety of mechanisms, including osmotic adjustment, hydraulic status, phytohormones, and ROS signals (Moreno-Galván et al., 2020). These ROS are extremely toxic to plant cells and can oxidise essential cellular components such as nucleic acids, fats, and proteins, promoting lipid peroxidation, protein denaturation, and DNA mutations, resulting in pigment complex degradation, cell death, and reduced plant biomass (Huang et al., 2019). While fresh and dried weights of DK-9108, DK-6321, and Sarhaab were enhanced in AB-amended soil, the greatest increase was observed in DK-6321 at four tonnes acre−1 AB, even at 50% and 70% IW. The AB amendment acts as a water bank, protecting the plat cell membrane from oxidative damage at the permanent wilting point by gradually providing moisture and nutrients to maintain normal water potential and achieve cell turgidity for maximum transpiration, photosynthesis, and stomatal conductance (Bu et al., 2020; Hafeez et al., 2017; Tayyab et al., 2018). Adding AB to soil increased carbon storage, nutrient retention, and water uptake around roots, which led to better plant growth in water-stressed situations (Romdhane et al., 2019; Safari et al., 2023).

The length of the shoots and roots decreased under water stress in all maize hybrids, particularly in Sarhaab, in soil that did not include any AB amendment. The reduction in water availability hindered the cell elongation mechanism, which was responsible for the root and shoot length decline. Water stress arrests the flow of sugars from the leaves to the growing regions of roots and shoots. It also stopped cells in the roots and shoots from stretching (Khan et al., 2019). Severe water stress affected soil characteristics and nutrient transfer from soil to plants, resulting in decreased root and shoot length due to a nutrient deficit even in fertilised soil (Bista et al., 2018; Seleiman et al., 2021). However, soil amendment with AB resulted in increased root and branch length in Sarhaab, DK-9108, and DK-6321. AB boosted soil moisture and critical nutrients, improving soil fertility and making it available to maize plants (Murtaza et al., 2021). The nutrients and moisture stored by AB help plants control their metabolic activity during drought stress (Bitarafan, Liu & Andreasen, 2020; Ali et al., 2021). The present research outcomes, which indicate elevated levels of photosynthetic pigments and enhanced relative water content, may additionally be associated with elongated roots and shoots in plants experiencing water scarcity.

In all maize hybrids, yield parameters such as ear length, ear weight, total number of grains per ear, and 1,000 grain weight were significantly reduced at 50% IW in soil without AB amendment, with the biggest decline observed in Sarhaab (most drought sensitive). Water stress alters plant yield metrics (Bodner, Nakhforoosh & Kaul, 2015). Water stress during grain filling in maize may result in reduced ear weight and length, total number of seeds per ear, and weight per thousand grains (Elshamarka et al., 2024). Water stress reduces yield parameters by reducing photosynthesis (due to reduced photosynthetic pigments) and hindering flower and grain filling, resulting in fewer ear per plant, fewer grains per ear, and lower grain weight (Seleiman et al., 2021; Yousaf et al., 2022). All maize hybrids planted in soil amended with 2-tonnes acre−1 and 4-tonnes acre−1 showed higher ear length, ear weight, total number of grains per ear, and 1,000 grain weight; DK-6321 showed the largest increase in 4-tonnes acre−1 soil even under 50%–70% IW. Because of their large surface area and pore volume, AB amendments improve the soil’s ability to retain water, allowing the soil to hold a suitable amount of plant-available water during times of stress (Niu et al., 2017; Singh, 2019). Hence, one of the best methods for lessening the detrimental effects of water stress on maize productivity is enhanced soil structure and water retention capacity via activated biochar.

Conclusion

Moisture stress is among the most severe abiotic factors impacting plant growth and productivity globally. This study demonstrates that water deficiency significantly impairs maize growth and yield. The application of activated biochar (AB) was found most efficient in enhancing the soil’s physical and chemical properties. The positive effect of biochar in reducing moisture stress was evident in the improved plant growth, demonstrated by the highest improvement in photosynthetic pigments, and reduced stress-related osmolytes like proline and soluble sugars in maize plants. Maize plant’s moisture stress tolerance was linked to the preservation of chlorophyll pigments for enhanced activity of the photosynthetic system, and improved regulation of cellular membranes, which together contributed to an increase in final grain yield. Our research highlights the significance of using vermicompost to organically activate biochar and shows how AB soil amendment can be utilized to boost maize yield in dry, arid regions. This research highlights the potential of organically activated biochar as a win-win strategy to mitigate moisture stress by retaining moisture and nutrients in agricultural soil for extended periods, benefiting plant growth under water-scarce conditions. Therefore, it is recommended that future research focus on assessing the performance of AB with different agricultural practices such as crop rotation and intercropping to improve soil health and crop productivity in arid agricultural systems. By integrating AB into agricultural policies, the adoption of sustainable farming practices that improve soil health and water retention can be encouraged, thereby reducing dependency on chemical fertilizers. However, farmers may require technical knowledge and expertise to effectively apply biochar. Without adequate training, the potential benefits of biochar might not be fully realized. Moreover, exploring the role of AB in carbon sequestration and its potential to mitigate climate change effects can provide an additional incentive for its widespread adoption. Detailed studies on the carbon footprint of biochar production and application are needed.

Supplemental Information

Supplemental Information 1 Raw data

Supplemental Information 2 Various stages of experiment from activation of biochar, soil preparation, sowing, vegetative growth, and harvesting of maize

Supplemental Information 3 Figures showing various stages of experimental procedures

Supplemental Information 4 Codes for Chord diagram and heatmap

We would like to acknowledge Bayer, Lahore, for providing the seeds of the maize hybrids.

Abbreviations

AB Activated biochar

OM Organic matter

TOM Total organic carbon

ETc Evapotranspiration of crop

DR Depth of root

FTIR Fourier Transform Infrared

IW Irrigation water

WHC Water holding capacity

ROS Reactive oxygen species

Additional Information and Declarations

Competing Interests

Author Contributions

Data Availability

The authors declare there are no competing interests.

Hajra Iqbal conceived and designed the experiments, performed the experiments, analyzed the data, prepared figures and/or tables, and approved the final draft.

Summera Jahan conceived and designed the experiments, performed the experiments, analyzed the data, prepared figures and/or tables, authored or reviewed drafts of the article, and approved the final draft.

Muhammad Akbar analyzed the data, prepared figures and/or tables, authored or reviewed drafts of the article, and approved the final draft.

Atif Kamran analyzed the data, authored or reviewed drafts of the article, and approved the final draft.

Fahd Rasul conceived and designed the experiments, performed the experiments, analyzed the data, authored or reviewed drafts of the article, and approved the final draft.

Sobia Shahzad performed the experiments, analyzed the data, authored or reviewed drafts of the article, and approved the final draft.

Manzer H. Siddiqui analyzed the data, authored or reviewed drafts of the article, and approved the final draft.

Saud Alamri analyzed the data, prepared figures and/or tables, and approved the final draft.

The following information was supplied regarding data availability:

Raw data are available in Supplemental Files.

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
