# Peer review of "Unveiling the impact of organically activated biochar on physiological, biochemical, and yield attributes of maize under varied field moisture conditions"

_PeerJ, doi:10.7717/peerj.17883_

## Round 0.1 · original submission · Major Revisions

Drought stress is a significant issue, and understanding how crops like maize respond to such adverse conditions is crucial for reducing its negative impact. The authors noted that their approach, using biochar, could help mitigate these effects on maize. However, it is essential to address certain technical details to refine your article. I strongly recommend thoroughly reviewing the reviewers' suggestions and carefully considering each recommendation. Additionally, you should submit the raw data from your research and the R codes used. If you disagree with specific suggestions, providing a clear and well-justified rationale for your perspective would be beneficial.

Reviewer 1 ·

Basic reporting

The manuscript is good and in the interest of the readership of the Journal, can be accepted for publication but the following improvements may be added in the revised version.
1. Abstract must be reframed, it should be a crisp summary of the study and it must reflect the separate entities, with a cut message

2. The introduction is ok, please add the study gap and also state your novelty.

3. Methodology must be expanded, kindly elaborate on the sampling technique and lab analysis

4. Results and discussion need to be updated with current citations from highly reputed journals.

5. Authors suggested adding on subhead about the policy implications of the study and also highlighting the constraints and futuristic scope of the work.

6. Please check the pattern and style of the journal, references must be cross-checked critically to improve the language of the article.

7. All the tables and figures must be self explanatory, Figure should have >300 dpi.

8. Conclusion must be generalized it should not be a repetition of results.

Experimental design

The experimental design is perfect and statistical analysis sound

Validity of the findings

Findings are good and reproducible

Additional comments

The manuscript can be accepted after major revision.

·

Basic reporting

Major comments:
The manuscript presents a comprehensive overview of the research focused on the impact of activated biochar (AB) amendment on maize crops under different water stress conditions in Punjab, Pakistan. The key findings indicate that the application of AB significantly improves soil physical and chemical properties, maize physiological and biochemical attributes, and yield parameters, particularly under drought stress. The scientific background is not sufficiently explained and there is a lack of understanding. Some of the literature is older and it is necessary to use the most recent literature, not older than 5 years. The literature review is not strong enough to detect gaps in the research. The discussion of the obtained results is not well explained and needs more comparison with the literature. Additionally, the limitations of the research and areas where further research is needed should be addressed. Also, the meaning of all abbreviations used in tables should be provided. The scientific English language used in the manuscript needs some improvement as there are punctuation and grammatical mistakes present throughout the manuscript. Sentences need more clarity and better construction. Some sentences and phrases could be rephrased or simplified to improve clarity. Moreover, there should be consistency in the representations of figures/graphs in terms of units, style, colour, font size, figure names, etc. Due to the listed reasons, the quality of the manuscript does not meet the standards of PeerJ or needs major revisions.

Introduction:
The introduction is too short and does not provide enough information about the study. The introduction needs to be strengthened in terms of recent research in this area with possible research gaps and future applications of this study. There is a lack of sufficient background information. The introduction needs to emphasize the research work with a detailed explanation of the whole process considering past, present, and future scope. Research gaps should be highlighted more clearly and future applications of this study should be added.

Specific comments:
1. Abstract: The abstract only contains some parameters without any process conditions or key values from results, which is insufficient to delineate the whole picture of the contribution and possible application of this study. It is suggested to add key values from the results, and possible applications of this study and highlight the novelty of this work clearly (200-250 words).
2. Revise keywords and add more specific and novel keywords with broader meanings (5-7 words).
3. Page numbers are missing in the manuscript.
4. The introduction and background are weak, no strong information is provided about the different types of the agricultural system and their effects on crops and wastewater and environmental pollution; therefore, the authors are advised to read, broader the literature review and add environmental pollution types, effects and levels from the following studies by extending literature
5. Page 8, line 157: “The surface area of biochar can be enhanced by its activation, which involves the improvement in its surface through organic or chemical treatments.” Which of the two mentioned treatments has proven to be more efficient?
6. Page 8. line 167: “Biochar production was undertaken in 2022 by utilizing raw material sourced from the gum Arabica tree (Acacia nilotica) at the University of Agriculture, Faisalabad.” Why was this specific tree chosen for biochar production? What properties make this tree suitable for this study?
7. Page 9, line 184: “The field experiment was conducted at location 32°38.37’N, 74°9.00’E (Gujrat) for exploring the suitable level of activated biochar under three moisture regimes (100% ETC, 70% ETC, and 50% ETC).” On which basis was this location chosen for conducting a field experiment?
8. Page 9, line 196: “The solution was centrifuged (5 minutes) and the upper layer was transferred to the clean cuvette of the spectrophotometer and absorbance was recorded at 663-nm, 645-nm and 470-nm.” Which RCF force was used during centrifugation?
9. The discussion presented is very weak no strong comparison has been made with the literature to support the authenticity of the obtained results. Therefore, the authors are suggested to discuss their results with the following recent researches about novel irrigation systems, water, nano fertilizers, their effect on production, soil properties and related pollution effects to make the background and discussion more strong
10. Page 14, line 368: “To cope with drought, plants evolved a variety of morphological, biochemical, and physiological adaptations.” Can you mention some of these adaptations?
11. Page 14, line 372: “However, by increasing its surface area and pore capacity, organic activation of the biochar further improves its ability to store water for longer periods of time.” Can you compare the durability of water storage when it comes to organic treatment versus some previously reported chemical treatment?
12. Page 16, line 447: “As a result, the leaves ultimately exhibited an increase in carbohydrate and amide concentrations (Gharred et al., 2022; Khan et al., 2021; Singh et al., 2020).” Avoid an abundance of references do not cite more than 2 references in a single place. Correct all these types of references throughout the manuscript.
13. The equations should be numbered and properly cross-referenced in the manuscript. Please correct this.
14. Revise figures in the manuscript. Draw all figures in high-quality figures should be coloured and attractive.
15. Please revise all the tables and figures captions for a better understanding. Should be comprehensive and meaningful.
16. Conclusion: The conclusions only talk about some studied parameters, which is insufficient to depict the whole picture of the contribution of this study. The authors are advised to write the conclusions in a comprehensive way and should contain key values, suitability of the applied method, the major findings, contributions and possible future outcomes (250-300 words).
17. References: The authors are advised to include the latest references. Please see some suggestions in the specific comments section. Further discussion should be modified with the latest references.
18. The authors are advised to include a list of abbreviations in the manuscript.

Experimental design

See comments in basic report.

Validity of the findings

See comments in basic report.

Additional comments

See comments in basic report.

Reviewer 3 ·

Basic reporting

no comment

Experimental design

- Information should be given about how irrigation is done

- Information such as how many cm between the rows and on the rows, how many corns are in each parcel should be given in this section

- The field trials information in the Materials and Methods section should be written more clearly.

Validity of the findings

no comment

Additional comments

The article may give many applicable knowledge to the scientist in terms of biochar activity. Although this topic is not a novel subject for for us, approach of the researchers to the problem is logical and attractive. I think the data from this article can use many scientist in many ways. I wrote my suggestion on it.

Annotated reviews are not available for download in order to protect the identity of reviewers who chose to remain anonymous.

---

## Round 0.2 · accepted · Accept

I appreciate your constructive attitude toward the suggestions of reviewers and improving your article based on their suggestions. I believe your manuscript is now ready for publication. We look forward to your next article.

Reviewer 1 ·

Basic reporting

Manuscript can be accepted in current form

Experimental design

Manuscript can be accepted in current form

Validity of the findings

Manuscript can be accepted in current form

Additional comments

Manuscript can be accepted in current form

·

Basic reporting

See additional comments

Experimental design

See additional comments

Validity of the findings

See additional comments

Additional comments

The authors have addressed most of the comments; they have also tried to make changes according to the reviewers’ suggestions. After revisions, the quality of the manuscript has been adequately enhanced. Therefore, the manuscript could be considered for publication in Energies Journal. However, there are still some editing/ syntax errors present in the manuscript which need to be corrected, hence the authors are advised to read the manuscript carefully during proofs.

Reviewer 3 ·

Basic reporting

No comment

Experimental design

No comment

Validity of the findings

No comment

Additional comments

The changes suggested in the article have been made. The article can be published in this form.